# Effect of Coffee on the Bioavailability of Sterols

**DOI:** 10.3390/foods11192935

**Published:** 2022-09-20

**Authors:** Cristiana L. Pires, Inês M. V. Silva, Manuel A. Coimbra, Maria João Moreno, Filipe Coreta-Gomes

**Affiliations:** 1Coimbra Chemistry Centre, Institute of Molecular Sciences (CQC-IMS), University of Coimbra, 3004-535 Coimbra, Portugal; 2Department of Chemistry, Faculty of Science and Technology, University of Coimbra, Rua Larga, 3004-535 Coimbra, Portugal; 3LAQV-REQUIMTE, Chemistry Department, University of Aveiro, 3810-193 Aveiro, Portugal

**Keywords:** cholesterol, bile salts, coffee, hypocholesterolemic, permeability, Caco-2 monolayers, dehydroergosterol

## Abstract

Absorption at the intestinal epithelium is a major determinant of cholesterol levels in the organism, influencing the entry of dietary cholesterol and the excretion of endogenous cholesterol. Several strategies are currently being followed to reduce cholesterol absorption, using both pharmacological agents or food ingredients with hypocholesterolemic properties. Coffee has recently been shown to affect cholesterol bioaccessibility, although it has not been shown if this translates into a decrease on cholesterol bioavailability. In this work, coffee obtained with different commercial roasting (light and dark) and grinding (finer and coarser) was evaluated regarding their effect on cholesterol absorption through Caco-2 monolayers, mimicking the intestinal epithelium. The fluorescent dehydroergosterol was used as a sterol model, which was shown to permeate Caco-2 monolayers with a low-to-moderate permeability coefficient depending on its concentration. In the presence of coffee extracts, a 50% decrease of the sterol permeability coefficient was observed, showing their potential to affect sterol bioavailability. This was attributed to an increased sterol precipitation and its deposition on the apical epithelial surface. A higher hypocholesterolemic effect was observed for the dark roasting and finer grinding, showing that the modulation of these technological processing parameters may produce coffees with optimized hypocholesterolemic activity.

## 1. Introduction

High levels of cholesterol in the blood have been associated with the development of cardiovascular diseases, which have a high mortality and disability indexes [1,2]. Cholesterol sources can be either endogenous, due to its synthesis by the liver, or exogenous, supplied by the diet, amounting a daily total of 1800 mg [3]. Several pharmacological and functional food hypocholesterolemic approaches are known to address both sources, namely: (a) inhibitors of the synthesis of endogenous cholesterol (e.g., statins) [4]; (b) inhibitors of the cholesterol transporter Niemann Peak C1 Like 1 (e.g., Ezetimibe) [5]; (c) cationic resins (e.g., Colesevelam) [6,7], (d) polysaccharides (e.g., beta-Glucans) sequestering bile salts, affecting cholesterol bioaccessibility [8,9]; and (e) phytosterols (e.g., functional yogurt’s and margarine) affecting cholesterol bioaccessibility [10,11]. These hypocholesterolemic strategies can be used alone or synergistically, being dependent on the origin and stage of the disease. Functional foods are usually used as a preventive approach in an early stage of the disease, and at later stages it complements pharmacological options for a more effective reduction of cholesterol levels. The use of functional foods may additionally allow a reduction in the dose of the pharmacological agent. If accurately combined, it can reduce the adverse secondary effects of drugs and increase the patient compliance to the treatment [12]. In order to be effective, functional food should be taken after major meals where a higher content of cholesterol from diet and from discharges of the gallbladder are present at the intestine. The emulsification of sterols depend on bile salts [13], which can be either synthesized endogenously or through the microbiota. The number and position of bile salt hydroxyl groups influence their capacity to emulsify cholesterol. At 50 mM bile salt concentration, the glycodeoxycholic acid (GDCA) with two hydroxyl groups shows the highest capacity to emulsify cholesterol (up to 3 mM) followed by glycochenodeoxycholic acid (GCDCA) (1.3 mM), the tri-hydroxy glycocholic acid (GCA) showing the smaller emulsifying capacity (1.0 mM) [10,14]. This reflects the concentration of BS micelles, which depends on the critical micelle concentration (CMC) that is smaller for the di-hydroxy BS (1.6 and 1.7 for GCDCA and GDCA, respectively) and larger for GCA (6.5 mM). The cholesterol emulsifying capacity is also dependent on the properties of the micelles formed, justifying the lower emulsifying capacity of GCDCA in spite of its lower CMC. Positive synergy in the cholesterol emulsifying capacity was previously observed for the BS mixture present in the human intestinal lumen (GCA/GCDCA/GDCA 37.5/37.5/25), which at 50 mM total BS are able to emulsify up to 1.8 mM cholesterol [10].

Currently, most of the hypocholesterolemic functional foods are based on dairy matrices enriched in phytosterols and cereals enriched in β-Glucans, which are able to affect the emulsification process, either competing with cholesterol for the solubilization in the bile salt micelle or by sequestrating bile salt from the solution leading to the precipitation of cholesterol [8,10]. The physico-chemical properties of these functional foods do not allow the inclusion in hydrophilic matrices, thus limiting their consumption and constraining the meals intake time. Food matrices compatible with consumption at major meals and integrated in diet habits would maximize their efficacy and acceptance. Coffee stands up as a very promising food matrix due to its generalized consumption, including the periods after major meals, and due to its capacity to decrease cholesterol bioaccessibility [9]. The mechanism of the hypocholesterolemic effect was attributed to the sequestration of bile salts by the polysaccharides present in coffee, namely by arabinogalactans and galactomannans. Moreover, lipids present in coffee can also co-solubilize in the dietary micelles decreasing cholesterol bioaccessibility [9]. Considering the commercial coffee roasting ranges, both polysaccharides and lipids increase their content in the brew with the increase of roasting degree [15]. Coffee roasting also leads to the formation of melanoidins, high molecular weight compounds resultant from Maillard reactions between polysaccharides, protein and phenolic compounds [16,17,18,19], which may have also a role in the hypocholesterolemic properties of coffee brews. Grinding can also have a role in these bioactive properties. Finer coffee particles allow for the extraction of more total solids, sugars and polysaccharides to the brews due to the higher surface area, which promotes higher liquid-solid extraction during the water percolation through the coffee particles [20]. An increase in grinding level usually enhances the extraction of compounds due to the increase of surface contact area with the extraction water solvent [21]. The main objective of the present work is to evaluate if the effect of coffee in decreasing cholesterol bioaccessibility using the in vitro intestinal model translates into a decrease in the absorption of cholesterol through the intestinal epithelium, an indication of their potential to affect sterol bioavailability. For this goal the effect of coffee in the permeability through Caco-2 monolayers was addressed. Those permeability assays are well-established in vitro procedures to evaluate absorption at the intestine [22], and to predict the bioavailability of poorly metabolized and slowly excreted compounds [23,24,25,26,27]. Cholesterol absorption at the intestine involves both passive and active mechanisms [5,28]. In this work the focus is on passive processes of cholesterol absorption. For that purpose the sterol ergosta-5,7,9(11),22-tetraen-3β-ol (DHE) is used [28,29]. This fluorescent sterol is a good model of cholesterol when passive processes are being addressed, and was shown to describe accurately cholesterol dynamics and equilibrium distribution in the blood [30]. The high fluorescent quantum yield of DHE increases the sensibility in its detection and permits the quantitative characterization of its apparent permeability coefficient through the Caco-2 monolayer. Additionally, this property allows the use of confocal microscopy to characterize sterol location and better understand the mechanism of coffee hypocholesterolemic effect. 

## 2. Materials and Methods

### 2.1. Reagents and Materials

Arabica Brazil coffee were gently supplied by the Portuguese coffee roaster company FEB (Taveiro, Coimbra, Portugal). 

The Caco-2 cell line was purchased from European Collection of Authenticated Cell Cultures (ECACC 09042001, Salisbury, UK). Dulbecco’s modified Eagle’s medium high glucose (DMEM, 4.5 g/L glucose with 2 mM L-glutamine), sodium bicarbonate, nonessential amino acids (NEAA), penicillin (10,000 units/mL)–streptomycin (10 mg/mL) antibiotic, 0.25% (*w*/*v*) trypsin, ethylenediamine tetraacetic acid (EDTA), Hank’s balanced salt solution (HBSS), 4-2-hydroxyethyl-1-piperazineethanesulfonic acid (HEPES), Lucifer yellow CH di-potassium salt (LY), Hoechst 33,342, dehydroergosterol (DHE) and the sodium salts of glycocholic acid (GCA), glycodeoxycholic acid (GDCA) and glycochenodeoxycholic acid (GCDCA) were all purchased from Merck S.A (Algés, Portugal). Fetal bovine serum (FBS) was obtained from Gibco-Life Technologies (Porto, Portugal) and bovine serum albumin (BSA) was from Applichem (Darmstadt, Germany). 3-(4,5-dimethylthiazol2-yl)-2,5-diphenyltetrazolium bromide (MTT) was acquired from Alfa Aesar (Kendal, Germany). Corning^®^ Transwell^TM^ 12-wells permeable polycarbonate membrane inserts (1.12 cm^2^ surface area, 0.4 μm pore), 12- and 96-wells cell culture plates were obtained from VWR (Amadora, Portugal).

Zonula occludens-1 (ZO-1) rabbit polyclonal primary antibody and Cy3 Goat anti-rabbit IgG secondary antibody used in immunostaining for confocal microscopy were purchased from Alfagene (Lisboa, Portugal). 

Methanol and acetonitrile HPLC-grade 99.9% purity were from Fisher Scientific (Lisboa, Portugal).

### 2.2. Preparation of the Coffee Extracts

Arabica Brazil green grains were subjected to a commercial roasting procedure in the company FEB, originating commercial light (L) and dark (D) roasted coffee beans. Using a coffee grinder, coffee beans were milled with two different grinding levels to obtain fine (1) and coarse (3) particle sizes. The final coffee beverages, denoted as L1 (light roast and fine grinding), D1, L3 and D3 were extracted in an expresso machine (Flama 10) operating at 15 bar in the proportion of 6 g per 40 mL of water. In order to have uniform extractions, a tamper of 51 mm was used to pressure the coffee powder in the coffee portafilter basket previous to the extraction. The sample corresponding to 50 beverages was lyophilized at −42 °C, 100 mT in a Kinetics Ez-Dry freeze dryer (Kinetics, Dresden, Germany) coupled to an Edwards 12 vacuum pump (Edwards, Burgess Hill, UK). After freeze-drying, the powders were weighed and stored at room temperature for further studies. The coffee concentration ranged from 32.7–34.9 mg/mL for the four samples, giving on average 1.0 g of coffee in 40 mL of an espresso like beverage.

### 2.3. Caco-2 Cell Culture

The Caco-2 cells (passages 95–105) were routinely cultured in DMEM medium supplemented with 1.5 g/L sodium bicarbonate, 10.0% (*v/v*) heat-inactivated FBS, 1.0% (*v/v*) NEAA and 1.0% (*v/v*) Pen-Strep. Cells were grown in 75 cm^2^ culture flasks at 37 °C under an air-humidified atmosphere of 5% CO_2_. When reaching 80–90% confluence, usually after 3–4 days, the cells were detached using trypsin-EDTA and subcultured at a 1:8 split ratio. Further details on the procedure followed are described in Pires et al. [31].

### 2.4. Cell Viability Assays

The influence of bile salts and coffee extracts on the Caco-2 cells viability was assessed with the MTT assay for determination of mitochondrial activity in active cells [32]. Briefly, the cells were seeded in 96-well plates at a density of 1 × 10^4^ cells/well in 200 μL of medium. Cells were then maintained in culture until confluence, for 96 h with medium renewal after 3 days. Solutions of individual bile salts GCDCA, GDCA, GCA and the combination of all three were prepared in the transport buffer HBSS (containing 25 mM HEPES, 0.35 g/L sodium bicarbonate, pH 7.4) at several concentrations above their critical micellar concentration (CMC). Two-fold serial dilutions of individual coffee extracts or mixtures with the bile salts were prepared by diluting the lyophilized extracts in HBSS or HBSS plus bile salts. The following controls were also included: 20% (*v*/*v*) of dimethyl sulfoxide (DMSO) in HBSS (positive control) and only HBSS (negative control). The cells were incubated for 6 h with test samples or controls (200 μL). Then, the cells were washed with PBS and incubated with 150 μL of MTT diluted in medium (0.5 mg/mL) for 4 h at 37 °C. Finally, the medium was discarded, and the formazan crystals produced by active cells were solubilized with dimethyl sulfoxide (150 μL) for 15 min under continuous stirring. The absorbances were measured at 570 nm in a Synergy HT Microplate Reader (BioTek Instruments, Inc., Winooski, VT, USA). All conditions were performed in triplicate. Results are expressed as a percentage of cell viability relative to the negative control.

### 2.5. Absorptive Permeability Assays

For transport experiments, the Caco-2 monolayers were prepared by seeding the cells at a density of 2.6 × 10^5^ cells/cm^2^ on 12-well polycarbonate Transwell^TM^ plates, following protocols previously defined for this assay [31,33]. The cells were cultured until monolayer differentiation, for a total of 24 days, with culture medium being renewed every two days. On the day of the experiment, the culture medium was removed and replaced by HBSS medium pre-warmed at 37 °C (0.5 mL in the apical and 1.5 mL in the basolateral sides). The cell monolayers were placed in an incubator at 37 °C with a plate orbital shaker (IKA-Schüttler MTS4) at 50 rpm for 8 min. After two washing steps, the transepithelial electrical resistance (TEER) values were measured using a Millicell^®^ ERS-2 voltmeter (Merck). The net TEER values (Ω cm^2^) were obtained by subtracting the cell-free filter resistance and multiplying by filter area. All cell monolayers achieved TEER values exceeding 700 Ω cm^2^ and were considered adequate for transport experiments. 

In assessing the influence of coffees samples on DHE absorption by intestinal epithelia, the transport was studied in the absorptive way; that is, in the apical (mimicking the intestinal lumen) to the basolateral (mimicking the bloodstream) direction. The solutions of DHE to be applied in the apical side were prepared as follows: first, the lyophilized coffee extracts (16-fold diluted) were dissolved with GCA (10 mM) solution in HBSS. Then, the mixtures were allowed to equilibrated overnight in a 37 °C in a water bath with shaking at 100 rpm. On the day of the experiment, DHE was added to the coffee/GCA mixture or to GCA only (Control) as an aliquot from a DMSO stock of 10 mM DHE. Addition was performed under vigorous vortex mixing to facilitate DHE interaction with the GCA micelles and prevent its precipitation. The final DHE concentration ranged from 10 to 100 μM, corresponding to ≤1% (*v/v*) DMSO. To allow evaluation of the cell monolayer integrity in each permeability assay, the paracellular marker Lucifer Yellow (LY) was also added to the final mixtures at a concentration of 20 μM. The solutions were maintained at 37 °C until application in the permeability assay, and where optically clear.

Cholesterol has a very low solubility in aqueous media [34], with an estimated critical aggregation concentration for the cholesterol analog DHE of 2.5 × 10^−8^ M [35]. It was therefore necessary to guarantee that the composition of the acceptor compartment in the permeability assay has an adequate ability to solubilize the sterol and allow its quantification. To approach the in vivo conditions in absorptive cholesterol transport, the acceptor compartment contained serum albumin at 100 µM, which corresponds to the concentration usually found in interstitial tissues [36]. Serum albumin binds the cholesterol analog DHE with low to moderate affinity, *K*_B_ = 5 × 10^4^ [37], and at the concentration used DHE solubility is increased by at least an order of magnitude. To decrease the adsorption of DHE to the plate well and to the basolateral membrane, the transport media added to the basolateral side contained additionally 1% (*v*/*v*) DMSO. The presence of BSA and DMSO lead to improved sink conditions and higher DHE recoveries. The solutions from the acceptor compartments in the permeability assay where lyophilized directly in the wells of the permeability assay. A small amount of water was then added to facilitate detachment from the plate and solubilize salts and BSA. Ice-cold methanol was then added to dissolve DHE and precipitate the serum albumin. The DHE recovery efficiency when following this procedure was at least 90% (for details see Appendix A).

To start the absorptive transport experiments, the HBSS on the apical side was removed and replaced with 450 μL of the test solution (emulsified DHE + LY, emulsified DHE + coffee + LY or LY solutions). After addition of the test solution, 50 μL was collected (corresponding to a time delay of about 2 min) and the inserts were moved to a 12-well plate containing 1.2 mL of the basolateral transport medium. The plate was kept at 37 °C under stirring at 50 rpm for a total period of 6 h, with the assays being performed with either a single point sampling or three-points with 2 h sampling intervals. During the three-point sampling assay, 25 μL of the apical compartment was withdrawn at 2 and 4 h and replaced with fresh HBSS. At each time point indicated, the insert was also moved to another 12-well plate containing fresh basolateral medium (transfer sampling). At the end of the experiment (t_6h_), the inserts were transferred into empty wells and 50 μL were collected from the apical compartment and the remaining solution was decanted and stored. The 12-well plates with the samples of the basolateral compartments were frozen (−18 °C) and lyophilized. The same procedure was also applied to the aliquots of the apical side. All samples were then stored in a humidity-controlled environment until analysis.

At the end of the permeability assay the cell monolayers were placed in HBSS (0.5 mL apical and 1.5 mL basolateral) and TEER values were measured. The HBSS from this step was also collected and frozen at −18 °C for further analysis. Trypsin/EDTA (100 μL) was added to the inserts for cell detachment and cells were scraped of the filter and lysed with 1 mL of MeOH:Millipore H_2_O (3:1) as described by Broeders et al. [38]. Cells suspensions were then centrifuged for 10 min at 1200 rpm and the supernatant was collected and frozen (−18 °C) to further determine the cellular uptake of DHE.

Some of the cell monolayers were also prepared for visualization by confocal microscopy following the protocols describe in Section 2.9. For these experiments, an additional insert with a cell monolayer subjected to the same procedures but not used in permeability assays was also included as a control.

### 2.6. Quantitative Analysis of LY

The paracellular permeability of each cell monolayer was assessed by the quantification of LY transport from the apical and basolateral sides. For the samples from the apical side, aliquots of 50 μL were withdrawn at the beginning and end of the permeation experiment. The samples were diluted 30-fold and 100 µL were transferred to a 96-wells plate with clear bottom and black sides (Thermo Scientific Nunc). For the samples from the basolateral side, aliquots of 100 µL were withdrawn from each well containing the transport media for the sampled time intervals and transferred directly to the 96-well plate. The fluorescence intensity was measured at λ_ex_: 435 nm and λ_em_: 560 nm using the SpectraMax iD5 Multi-Mode Microplate Reader (Molecular Devices Corporation Sunnyvale, CA, USA). After the measurements, the solutions from the basolateral compartments were transferred back to their corresponding 12-well plate to be frozen and later analyzed for their DHE content. 

Quantification of LY was performed using calibration curves prepared in HBSS alone or HBSS with 100 μM of BSA, for samples from the apical and basolateral compartments respectively (see Appendix A). The parameters obtained for the validation of the analytical method, such as the linearity and the limit of quantification (MLQ) and detection (MLD), are also presented in Appendix A.

### 2.7. Quantitative Analysis of DHE

The DHE concentration in the samples was determined using a quaternary HPLC system (Agilent 1200 series G) equipped with a FLD detector, quaternary pump, and auto sampler. The column used was a Zorbax ODS C18 (250 × 4.6 mm, 5 μm), preceded by a pre-column with the same stationary phase and 12.5 mm length, both equilibrated at 30 °C. The detection was carried out at λ_ex_: 324 nm and λ_em_: 372 nm. The mobile phase used was 100% MEOH at a flow rate of 1 mL/min and the injection volume was 900 μL, with DHE showing a retention time of 12 min (Appendix A). Two sets of calibration curves were performed, using DHE solutions prepared in H_2_O:MEOH (1:6) (further details in Appendix A). Data acquisition and integration of peak areas was done using the Chemstation software B.03.01 from Agilent (Lisbon, Portugal). 

For the quantification of DHE in the samples from the permeability assays, DHE was extracted from the lyophilized samples using a mixture of methanol:water. The samples from the basolateral compartments were first resuspended in 190 μL of ultra-pure water (to dissolve the HBSS salts and albumin), and then 1140 μL of ice-cold methanol was added to precipitate the albumin and dissolve DHE (final ratio of water:methanol equal to 1:6). The samples were transferred to vials, mixed for 10 s and centrifuged at 1200 rpm for 10 min. The supernatant was collected and analyzed by HPLC for DHE quantification. The same procedure was applied to the lyophilized samples from the apical compartment. 

To assess the efficiency regarding DHE extraction, the same procedure was followed for control samples with known DHE concentration in the absence and presence of BSA and coffee extracts. Those assays were performed in triplicate, leading to a recovery of at least 90% in all conditions evaluated (for details see Appendix A). In the optimization of the procedure for DHE extraction, it was found that the addition of the small volume of water to the lyophilized powder before the addition of methanol was essential to obtain a good and reproducible extraction yield (results not shown). 

The quantification of DHE in the Caco-2 monolayers was performed directly on the supernatant obtained after centrifugation of the Caco-2 cells scrapped from the inserts (see Section 2.5 for details in this procedure).

### 2.8. Evaluation of the Apparent Permeability Coefficient

The apparent permeability coefficient (*P*_app_) of LY was calculated for each of the cell monolayers used to evaluate cell monolayer integrity. *P*_app_ was calculated from Equation (1) [31,33], and reported in cm/s.
(1)$$P_{\mathrm{app}}=\frac{\frac{\Delta Q^{A}}{\Delta t}V^{D}}{AQ_{0}^{D}}$$

In Equation (1), $\Delta Q^{A}$ is the amount of solute (mol) that accumulates in the basolateral compartment during the time interval Δt (in seconds), *A* is the surface area of the filter (1.12 cm^2^), $V^{D}$ is the volume of the donor (apical) compartment (0.4 cm^3^) and $Q_{0}^{D}$ is the initial amount of solute in the donor compartment (mol).

The same equation was used to calculate DHE *P*_app_ in the absence and presence of the coffee extracts.

### 2.9. Confocal Fluorescence Microscopy and Image Analysis

Following the permeability assay, some selected cell monolayers were prepared for analysis by confocal microscopy. First, the cell monolayers in the inserts were fixed with 4% of paraformaldehyde for 10 min at room temperature. After washing 3 times with phosphate-buffered saline solution (PBS), the inserts were then divided into two sets to be subjected to different analysis. 

The first set was stained for the visualization of the tight junction protein zonula occludens 1 (ZO-1) and nuclei. The protocol followed for immunostaining is described in detail in [31]. In brief, the cell monolayers were permeabilized with 0.1% (*w/v*) Triton X-100 in PBS for 10 min, washed again with PBS and blocked with 1% (*w/v*) BSA in PBS for 30 min at room temperature. Incubation with the primary antibody, a rabbit polyclonal ZO-1 antibody (diluted 1:200 with 1% (*w/v*) BSA), was performed at 4 °C overnight. The secondary Cy3 labeled goat anti-rabbit IgG antibody (diluted 1:100 with 1% (*w/v*) BSA) was added on the next day and incubated for 30 min at room temperature. Nuclei were stained with Hoechst 334,320 (1 μg/mL in 1% (*w/v*) BSA). In the second set, the cell monolayers were not stained or subject to any treatment after fixation.

Both the filter sets were cut, mounted on glass slides, covered with Dako mounting medium and stored at 4 °C in the dark until analysis. Images of the slides were captured with a Zeiss LSM 710, Axio Observer inverted confocal microscope at 20× magnification. For each condition, an average of 10 z-stacks were acquired with a slice distance of 1 μm. For the labeled cell monolayers, the following two filters’ settings were used to collect the confocal fluorescence images: ex: 405 nm or 561 nm with emission detection at 415–484 nm or 577–735 nm, respectively. For unlabeled cell monolayers, the samples were excited at 405 nm and the emission was collected at 408–490 nm.

Images post processing and analysis was performed using ImageJ software (version 1.8, NIH, Bethesda, MD, USA). First, images were cropped to an area of 108 × 108 µm^2^. Orthogonal projections of the XZ-plane were created from all the z-slices to view in detail the localization of the fluorescence in the cell monolayers.

### 2.10. Statistical Analysis

Statistically significant cytotoxicity differences between MTT controls and samples were tested using one-way ANOVA followed by a post hoc Dunnett’s test. For the permeability assays data, the statistical significance of differences between two values were determined using the unpaired Student’s *t*-test and in multiple comparisons one-way ANOVA was applied. GraphPad Prism software version 8.4.2 (San Diego, CA, USA) was used for the statistical analyses. Statistically significant differences were evaluated at 95% confidence interval.

## 3. Results and Discussion

### 3.1. Study of the Effect of Bile Salt and Coffee Composition and Concentration on the Intestinal Epithelium Model

In order to evaluate sterol bioavailability its permeability through a Caco-2 monolayer was evaluated. When grown for 21 to 28 days on semi-permeable filters, a tight monolayer of differentiated cells is formed, which mimics the intestinal epithelium [39]. The transport media was adjusted to mimic the in vivo system. The content of the donor (top compartment contacting with the apical surface of the cell monolayer) contained bile salts, sterol and coffee to model the intestinal lumen, while that of the acceptor (bottom compartment, in contact with the basolateral cell’s surface) contained serum albumin to model interstitial fluids. 

#### 3.1.1. Effect of Bile Salt Composition and Concentration on Caco-2 Monolayers Viability

Bile salts, as biological detergents used in vivo to emulsify cholesterol and other lipophilic substances, were selected for use to simulate the transport media in contact with apical side of the Caco-2 monolayers. The potential Caco-2 cells toxicity of the BS solutions prepared in HBSS buffer was evaluated at concentrations in the range 1.56–50 mM. Both primary bile salts (endogenously synthesized), glycocholic acid (GCA) and glycochenodeoxycholic acid (GCDCA), and the secondary bile salt (produced by microbiota), glycodeoxycholic acid (GDCA), were tested due to their abundance in the intestinal duodenum of humans. Furthermore, the most physiological relevant combination of the three bile salts (GCA/GCDCA/GDCA 37.5/37.5/25) was also tested at 3 and 6 mM. The results obtained for cell viability at all conditions tested are provided in Appendix A. Overall, the bile salt GDCA showed to be extremely toxic at all tested concentrations (<10% cell viability), while GCA exhibited no cytotoxic effects at concentrations ≤12.5 mM (>90% cell viability). This may be due to the much higher partition coefficient of GDCA for the cell membranes and its slower intrinsic rate of membrane permeation, leading to a very high local concentration and extensive perturbation of the membrane outer leaflet [40]. The mixed BS micelles were less toxic compared to pure GDCA but still exhibited severe toxic effects on the cells (<40% cell viability). The results obtained are in agreement with several studies that show that bile salts are cytotoxic in a concentration-dependent manner, and that this effect is strongly correlated with their hydrophobicity [41].

Given the lower toxicity of GCA, this was the selected bile salt and a more detailed study was performed to evaluate the maximal concentration that could be used in the permeability assays. The exposure of Caco-2 cells during 6 h to GCA at concentrations from 6.25 mM to 12 mM does not lead to a significant decrease in cell viability (always higher than 90%), while at 15 mM GCA cell viability is decreased to 80% when compared to the negative control (HBSS), Figure 1.

To maximize the solubilization capacity and therefore the sensitivity of the permeability assay of the transport media while maintaining a low toxicity, 10 mM was selected as the working GCA concentration in the donor (apical) side. At this concentration GCA is able to emulsify up to 10 µM DHE (Appendix A). At higher concentrations (up to 100 µM), the solutions are optically clear and stable for several hours, indicating that the aggregates of DHE formed do not interfere with the amount of DHE in the suspension. 

#### 3.1.2. Effect of Coffee Composition and Concentration on Caco-2 Monolayers Viability

Caco-2 cell viability assays were carried out to evaluate the influence of sample processing, as well as to determine non-toxic concentrations of coffees. Four processing conditions were used, namely, two roasting procedures (Dark, D and Light, L), followed by two grinding degrees (Fine, 1 and Coarse, 3). Stocks solutions were prepared from freeze-dried coffee extracts in HBSS buffer (33–35 mg/mL), which were then subsequently diluted by a factor of two (from 4 to 64-fold). The relevant cell viability results obtained with coffee L3 after an exposure period of 6 h are presented in Figure 2. 

Dilutions of the brew lower than 32 times showed cytotoxic effects (<80% cell viability) when compared with incubation with HBSS, while at 32-fold (1 mg/mL) or higher dilution no significant effects were observed on the viability of Caco-2 cells (94 ± 11%). Although some coffee extracts showed toxicity at 0.5 mg/mL, the majority do not present cytotoxicity at 1 mg/mL [42], in agreement with results obtained in this work This dilution may reflect the in vivo scenario when an expresso coffee is taken after a major meal. An empty stomach contains around 100 mL of fluid [30], which would lead to a dilution of around four times of an expresso coffee. However, coffee is usually consumed after meals, where the stomach volume may increase to above 1 L, leading to a dilution factor of more than 20 only in the stomach. 

Coffee toxicity was also evaluated in the presence of 10 mM of the bile salt GCA, to mimic the dietary intestinal content. The presence of GCA leads to a decrease in coffee toxicity to the Caco-2 cells at all coffee dilutions tested (>90% cell viability, *p* > 0.05, Figure 2). These results show a protective effect of GCA against coffee toxicity at high concentrations. Similar results were obtained with coffee extracts with distinct roast and grinding profiles, namely D1, D3 and L1 (Appendix A). Considering the cell viability observed, the dilution of 16-fold was selected as the working concentration to use in the permeability assays.

### 3.2. Effect of Emulsified DHE and Coffee on Cell Monolayer Integrity

The integrity of the Caco-2 monolayers after incubation for 6 h with the solutions containing bile salts, DHE and coffee was assessed by the use of three parameters: transepithelial electrical resistance (TEER) measurements, permeability of the paracellular marker Lucifer Yellow (LY) and the distribution of the peripheral membrane protein responsible for the organization of the tight junctions ZO-1.

Before the permeability assay, the average cell monolayer TEER values were 1723 ± 579 Ω cm^2^, being always higher than 700 Ω cm^2^. As the commonly accepted TEER value for a confluent and tight cell monolayer is 200 Ω cm^2^ [31,43], these were considered suitable to be used in the permeability assay. Following the exposure to samples for 6 h, the average TEER values of cell monolayers decreased to 720 ± 425 Ω cm^2^, maintaining values still above the accepted threshold. The decrease observed in TEER values after a permeability assay has been reported by several authors [31,44,45]. To complement the assessment of cell monolayer integrity by a different methodology, the permeability of a paracellular marker, LY, was used [31]. LY was added to the solutions of emulsified DHE alone and in the presence of the coffee extracts. Also, a solution with only LY was applied in a dedicated cell monolayer from the same Transwell^TM^ plate. After 6 h of exposure to the Caco-2 monolayers, the amount of LY that reached the basolateral compartment and the corresponding *P*_app_ values were calculated. 

The results shown on Table 1 were obtained with 10 µM DHE, close to the saturation of GCA micelles (Appendix A), in the absence and presence of the coffee extracts. DHE emulsified in GCA micelles in the donor compartment does not lead to an increase in the amount of LY transported, nor does the presence of the different coffee brews (Table 1). It is also observed that successive 2 h incubations do not lead to systematic variations in the % of LY transported. Altogether these results support the previous observations of nontoxicity of DHE, GCA or the coffee extracts at the concentrations used in the permeability assays. The permeability coefficient (P_app_) calculated (first 2 h) varied from 0.19 × 10^−6^ cm/s for L3 coffee to 0.42 × 10^−6^ cm/s for L1. These values are lower than the 0.5 × 10^−6^ cm/s value which is indicative of cell monolayer integrity [33]. The observed LY permeability is slightly higher (0.64 × 10^−6^ cm/s), although still within the expected result for cell monolayers on day 24 after seeding [31].

To evaluate the effect of DHE concentration on Caco-2 monolayer integrity, its concentration was increased to 50 and 100 µM. The results obtained for cell monolayer integrity are shown in Table 2. No significant effect was observed on LY permeability (average value of 0.8% for the amount of LY transported over the 6 h incubation, and 0.13 × 10^−6^ cm s^−1^ for *P*_app_), showing that possible DHE crystals formed did not influence the cell monolayer integrity. 

Overall, the results indicate that DHE, GCA, the coffee extract and samplings time points were not affecting the LY permeability in each of the cell monolayers used.

A third parameter was used to assess the cell monolayer integrity, the presence of the peripheral membrane tight junction protein ZO-1, visualized by immunofluorescence in a confocal microscope. The localization of the ZO-1 was evaluated in cell monolayers previously used in permeability assays where the apical compartment contained only GCA (10 mM), GCA + DHE (50 μM), GCA + coffee D1 (dil 16) and the mixture of the three components. Figure 3 shows the representative images of the z-stack projections obtained for cell monolayers used at each condition. At the end of the permeability assays, the cell monolayers show an extensive network of ZO-1, which is located only at the apical surface of the cells. The staining intensity and distribution is very similar to that observed in control cells, showing that the tested solutions do not influence the localization of ZO-1.

Considering all the results from the three parameters evaluated, it is concluded that the cell monolayers retained suitable integrity after the permeability assays with emulsified DHE alone or together with the coffee extract.

### 3.3. Influence of Coffee in DHE Bioaccessibility and Bioavailability

To evaluate the effect of coffee on cholesterol transport through the intestinal epithelium, DHE emulsified in GCA in the absence and in the presence of the four coffee brews was added to the apical side of Caco-2 monolayers. The amount of DHE in suspension in the apical compartment was quantified over time, with sampling immediately after addition (corresponding to about 2 min incubation), and at 2, 4 and 6 h. The results are shown in Figure 4, where the control corresponds to samples with DHE and GCA only. The amount of DHE is represented as % from that obtained for the sample before addition to the cell monolayer. 

A significant fraction of DHE (18.5 ± 3.9%) disappears from the transport media immediately after addition to the cell monolayers (Figure 4). The decrease was somewhat smaller (8.0 ± 3.1%) for the samples containing coffee independently of the brew considered. The decrease observed in the amount of DHE may be due to adsorption to the assay apparatus or to interaction with the cell monolayer, this being inhibited by the presence of the coffee extracts. Over the time course of the permeability assay, the amount of DHE in suspension in the apical compartment continued to decrease, with only 35 to 50% remaining after 2 h of incubation. Small additional decreases of only about 2% were observed at 4 and 6 h incubation. The logarithm time scale and the approximately linear relation shown suggest a simple monophasic process for disappearance with a slow kinetics. The process is not significantly affected by the presence of the coffee extracts, except for sample D1 (dark roasted and finer grinding), where a larger decrease is observed at 2, 4, and 6 h. The roasting and grinding of coffee promote the extraction of polysaccharides [15,20]. Considering that these high molecular weight molecules may bind bile salts [9], this could be a reason by which sterol bioaccessibility is being affected differently by these coffee extracts, although other compounds may also be responsible for this effect. 

To further understand the kinetics of DHE absorption to the cell monolayer and the effect of the coffee extracts, the appearance of DHE in the basolateral medium was also evaluated. The transfer method was used, the acceptor compartment being replaced by fresh medium at 2, 4 and 6 h. However, in spite of the large decrease observed in the amount of DHE in the donor (apical) compartment, DHE was not detected in the acceptor (basolateral) compartment. The detection limit of the analytical method used is equal to 1.6 pmol, which corresponds to 0.016% of the DHE added to the apical compartment. The low amount of DHE in the basolateral compartment may be the result of several processes, namely: (i) DHE is being sequestered by the cells; (ii) the 2 h sampling interval is too short for DHE to cross the cell monolayer; or (iii) DHE is being adsorbed to the assay apparatus (apical and basolateral containers or the polycarbonate membrane). 

To get insight regarding DHE location, the amount of DHE retained by the cell monolayer was evaluated at the end of the permeability assay. The cell monolayer was washed with HBSS and detached from the filter. Cells were then lysed with methanol:water 3:1 which also serves to extract DHE that was then quantified by HPLC. The results are shown in Figure 5A, with the amount of DHE extracted from the cells ranging from 16.3 ± 4.7 (control) to 27.5 ± 3.2% (coffee extract with D1 brew). The mass balance obtained for DHE at the end of assay is presented in Figure 5B. The recovery of DHE ranged from 56 to 72%, being similar for all samples. The higher amount of DHE in the cells obtained for coffee D1 agrees with the larger decrease observed for DHE in the apical compartment (Figure 4). These results indicate that coffee brews, dependent on their preparation, may have a role on the promotion of sterol intake by the intestinal epithelial cells. 

To calculate the DHE permeability coefficient it is necessary to have a measurable quantity of DHE in the basolateral compartment. To fulfill this requirement, the concentration of DHE in the apical side was increased as well as the incubation time. Higher DHE concentrations are in fact closer to the in vivo conditions when coffee is taken after a regular meal, where a high concentration of sterols intake is usually observed, leading to the saturation of bile salt emulsifying capacity [46]. Previous studies using model membranes have shown that the characteristic rate constant for desorption from membranes in liquid disordered and liquid ordered phases is equal to 1.0 × 10^−3^ s^−1^ and 6.9 × 10^−5^ s^−1^, respectively [37]. That is, the time required for 50% of the DHE molecules to desorb from the membrane should be within 0.27 to 4.0 h. Considering that the cell plasma membrane is enriched in cholesterol and saturated phospholipids [47,48], particularly high for endothelial and epithelial cell [49], those membranes are expected to have a high proportion of liquid ordered phases [50,51]. Therefore, a 6 h incubation time was used to guarantee that DHE desorb from the cell membranes and reach the acceptor compartment. Once in this compartment, DHE is sequestered by the BSA present in the transport media, leading to sink conditions in the permeability assay. As coffee D1 was the most effective in the transport from apical to the cell monolayer, this sample was used for the evaluation of DHE permeability coefficient, using concentrations of DHE in the apical side of the monolayer of 50 and 100 μM. 

### 3.4. DHE Bioaccessibility and Bioavailability at High Concentrations

The use of 50 and 100 μM of DHE allowed for the quantification of this compound in the apical, cell monolayer and basolateral compartments (Figure 6). The amount of DHE in the donor compartment was evaluated immediately after addition (represented by t_0_) and at the end of the permeability assay (t_6_), and is expressed as % of the total amount in the solution added (Figure 6A). A significant decrease in the percentage of DHE immediately after its addition to the apical compartment is observed. This decrease was higher for 100 µM, suggesting a more efficient transfer to the cell monolayer. After 6 h, the percentage of DHE that remains in the apical compartment is higher for the highest concentration. The increase of DHE concentration in the donor compartment may lead to the saturation of the cell’s monolayer. Considering that the cells have an average size of 20 µm × 20 µm and 8 µm thickness (Figure 3), and a 10-fold area increase of the apical side due to the presence of *villi*, the surface area available to interact with DHE is estimated as 5040 µm^2^ per cell, with a total plasma membrane surface of 1.4 × 10^9^ µm^2^ for the cell monolayer. This corresponds to about 5 nmol of lipids in the plasma membrane (Appendix A). Taking into consideration that a DHE concentration of 100 µM in the donor compartment corresponds to 20 nmol, saturation of the cell membrane with DHE seems a very plausible scenario. 

In the presence of coffee, the percentage of DHE immediately after its addition to the apical compartment remained higher than the controls. However, after 6 h incubation, no significant differences are observed relatively to the controls. This can be due to the retention of DHE by the coffee D1 sample in the apical compartment that slows down DHE transfer to the cell monolayer. 

Using 10 µM of DHE it was observed that 20% (corresponding to 0.8 nmol) was in the cell monolayer (Figure 6B). With 50 µM DHE, the percentage was 15% (corresponding to almost 3 nmol), approaching the estimated membrane saturation (50% sterol content in the lipid bilayer [49,52]). With 10 µM DHE, in the presence of coffee, the amount of DHE present in the cell monolayer was slightly higher than the respective control. However, this effect was not noticed with 50 µM DHE. 

The amount of DHE transported through the cell monolayer into the acceptor compartment is shown in Figure 6C. The amount of DHE transported increased with the increase in DHE concentration, leading to concentrations above the detection sensitivity for the case of 50 and 100 µM DHE. A significant decrease is observed in the amount of DHE transported through the cell monolayer when in the presence of the coffee extract. Those results show that coffee is effective in reducing the sterol permeability through the intestinal epithelium, leading to an expected significant decrease in the bioavailability of sterols.

From the amount of DHE that reaches the acceptor compartment one may calculate DHE apparent permeability coefficient (*P*_app_), Equation (1). In the absence of coffee, the values obtained for *P*_app_ range from 1.7 × 10^−9^ cm/s (for 50 µM DHE) to 2.6 × 10^−8^ cm/s (for 100 µM DHE), with the values of *P*_app_ being decreased by 50% in the presence of the coffee extract. However, some uncertainty in *P*_app_ measured may be attributed to the low DHE recovery [33] which was between 50 and 70%. The calculated *P*_app_ obtained show that DHE permeability through the Caco-2 monolayer is significantly lower than previously measured for cholesterol [53,54]. This difference may be due to several factors: (i) after accumulation of cholesterol in the plasma cell membrane, it reaches the endoplasmic reticulum where it is esterified and assembled in lipoproteins that are then exported to the basolateral media [55], sterols other than cholesterol are not esterified as efficiently [56,57], and therefore this transport pathway is not significant for DHE; (ii) the concentrations of sterol used in the previous works are significantly larger (up to 5 mM [54]), leading to a higher saturation of the cell membranes with an increase in the rate of sterol desorption from the membranes [52]; and (iii) when following cholesterol permeation, the sterol added to the donor compartment does not need to reach the acceptor compartment. Instead, cholesterol in the acceptor compartment is originated from the cell cholesterol pool. In contrast, because cells do not have DHE this sterol needs to equilibrate with all cell membranes before a significant accumulation in the basolateral compartment is observed. The permeability coefficient obtained in this work for DHE is thus a reporter of the rate at which sterols cross a cell monolayer by passive processes. Consequently, the significant decrease observed in DHE permeability in the presence of the coffee extracts is due to an effect on permeation by passive pathways. The work reported in this manuscript defines the basal line for passive process, allowing a better and quantitative assessment of active processes when both contribute to cholesterol permeability.

### 3.5. Localization of DHE in the Caco-2 Monolayers

To better understand the high amount of DHE associated with the Caco-2 monolayers and the effect of coffee on the DHE bioavailability, the cell monolayers incubated with 100 µM DHE were visualized by confocal microscopy. Representative images obtained after the permeability assays with DHE alone or DHE with coffee D1 are displayed in Figure 7. The insert with a control cell monolayer (not previously used in permeability assays) is shown in panel A, with the cells displaying some autofluorescence in the UV region dispersed throughout the cells surface. A small increase in fluorescence is observed for cell monolayers incubated with DHE (panel B) with the additional presence of some circular spots with higher fluorescence intensity. The density and intensity of the bright spots is significantly increased for the cell monolayers incubated with DHE and coffee (panel C). The lower plots in Figure 7 represent the z-stacks of the cross-section correspondent to the yellow lines in the main panels. From those images it is observed that the bright spots are located on the top of the cell monolayer apical membrane. No significant contribution of the coffee extract to the fluorescence intensity was observed (Appendix A), indicating that the bright spots observed in the presence of DHE and coffee are originated from DHE. The results obtained show that a significant fraction of DHE associated with the cell monolayer is not internalized by the cells. Instead, DHE is present as small aggregates adsorbed to the cell apical surface or to the coffee, namely its polysaccharides [9]. This explains why the presence of the coffee extract leads to a significant decrease in the amount of DHE transported through the cell monolayer (Figure 7C) in spite of the similar amount of DHE associated with the cell monolayer (Figure 7B).

In the preparation of the cell monolayer for confocal microscopy and quantification of DHE, the inset was pre-washed with buffer. This step may lead to the removal of sedimented material weakly adsorbed to the cell surface. In fact, a significant amount of DHE was quantified in the washing solution for the cell monolayers analyzed by confocal microscopy (26 ± 11% of the total DHE). This partially explains the small recovery observed when only the DHE associated with the cell monolayer, and that in the apical and in the basolateral compartments were considered.

## 4. Conclusions

In this work, the passive transcellular permeation of the fluorescent sterol DHE through Caco-2 monolayers was used to study the influence of different commercial roasted coffee extracts on sterol absorption. DHE solubility decreases in the presence of coffee extracts, this effect being higher for dark roasted and finer grinding. A decrease to about half is observed in the permeability coefficient of DHE in the presence of this coffee. This decrease is particularly relevant when considering that coffee is usually consumed after meals with a high cholesterol content. The consumption of coffee will therefore counterbalance the expected increase in cholesterol permeability due to the saturation of cell membranes. 

The results obtained by confocal microscopy for the cell monolayers in the presence of DHE and coffee extracts indicate that the decrease observed in the passive transcellular permeability of DHE is due to its precipitation in the donor compartment (mimicking the intestinal lumen). This agrees with the results obtained for the effect of coffee on cholesterol solubility in bile salt micelles, followed by NMR [9,14]. Both approaches reflect the effect of coffee on passive mechanisms of sterol absorption. To further disclose the hypocholesterolemic potential of coffee, the effect of its components on active mechanisms of cholesterol absorption and metabolism should also be evaluated.

## Figures and Tables

**Figure 1 foods-11-02935-f001:**
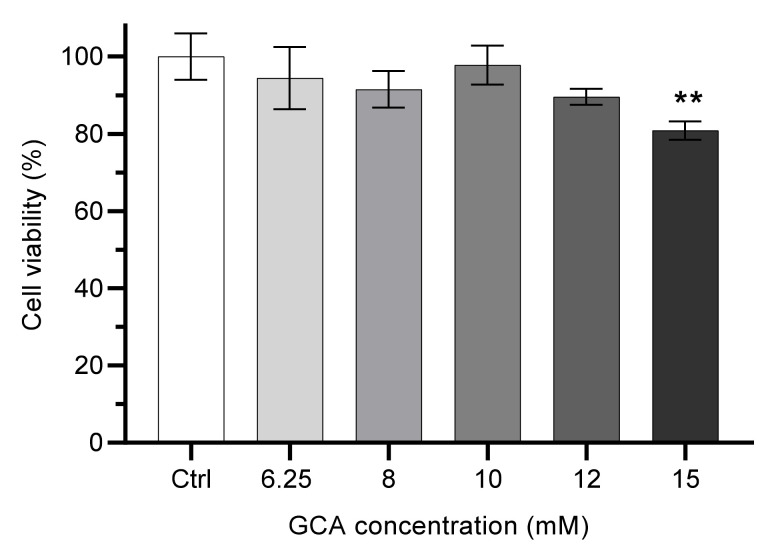
Effect of the concentration of bile salt GCA on Caco-2 cells viability. Cells were incubated with distinct GCA concentrations for 6 h. The GCA solutions were diluted with HBSS. Control cells were incubated in HBSS (100% viability). Statistically significant differences between the negative control are represented as ** *p* < 0.01, n ≥ 3.

**Figure 2 foods-11-02935-f002:**
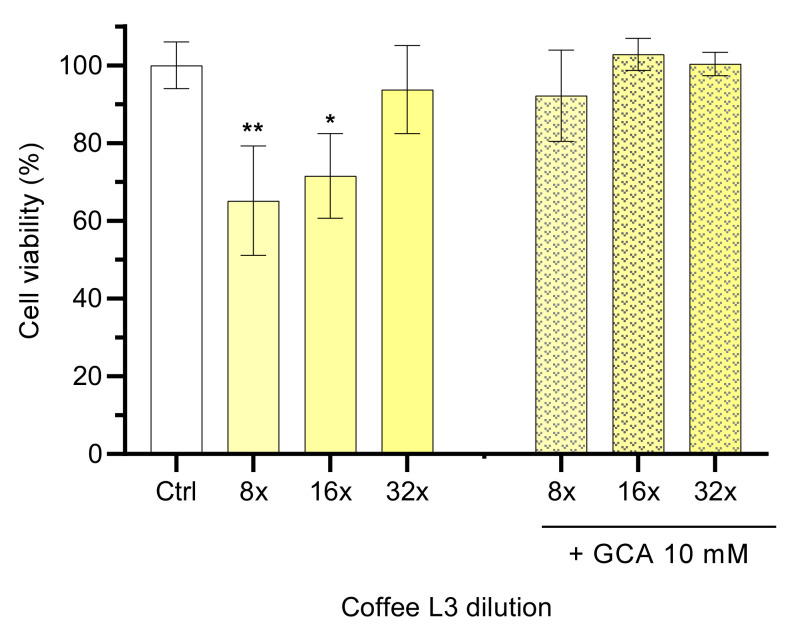
Effect of the light roasting with a coarse level coffee extract (L3) and its mixtures with GCA 10 mM on the viability of Caco-2 cells. The cell viability results are shown for coffee diluted 8- to 32-fold by a factor of four after cell exposure for 6 h. Dilutions were prepared with HBSS or GCA 10 mM in HBSS. Statistically significant differences between the negative control (HBSS) are represented as * *p* < 0.05 ** *p* < 0.01, n ≥ 3.

**Figure 3 foods-11-02935-f003:**
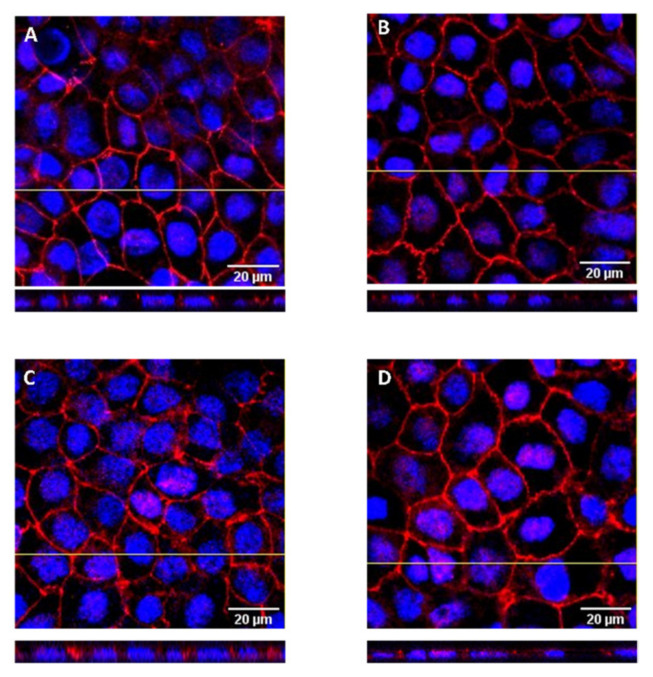
Confocal fluorescence images of Caco-2 monolayers stained for nuclei (blue) and ZO-1 (red). Representative optical section images along the *z*-axis (1 µm) are shown for cell monolayers after exposure during the permeability assay with: (**A**) GCA 10 mM, (**B**) DHE 50 µM + GCA 10 mM, (**C**) Coffee D1 + GCA 10 mM and (**D**) DHE 50 µM + coffee D1 + GCA 10 mM. The lower insets show the localization of the ZO-1 fluorescence in the orthogonal view of the z-x plane marked with a yellow line.

**Figure 4 foods-11-02935-f004:**
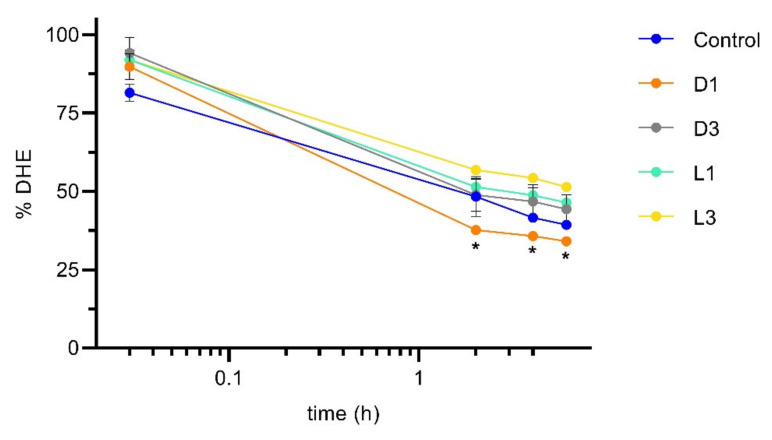
Amount of DHE in suspension in the donor (apical) compartment in permeability assays using Caco-2 monolayers. The sterol DHE (10 μM) was emulsified by GCA (10 mM) in the absence (control) and in the presence of distinct coffee brews (expresso like coffee diluted 16 times). The amount of DHE is given as % of that in the sample before addition to the cells. Statistically significant differences from the control are indicated by * (*p* < 0.05, n = 2).

**Figure 5 foods-11-02935-f005:**
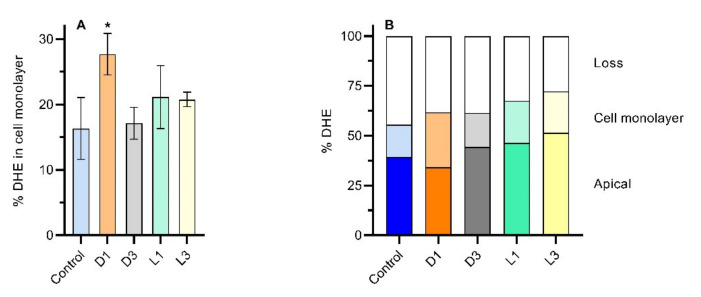
Quantification of DHE at the end of the permeability assays (6 h) for an initial concentration of DHE in the apical compartment equal to 10 μM. The amount of DHE extracted from the cell monolayer is shown in plot (**A**), and the distribution of DHE in the different compartments is shown in plot (**B**). The bars in dark colors represent DHE in the apical compartment, and those in light colors DHE extracted from the cell monolayer. Statistically significant differences from the control are indicated by * (*p* < 0.05, n = 2).

**Figure 6 foods-11-02935-f006:**
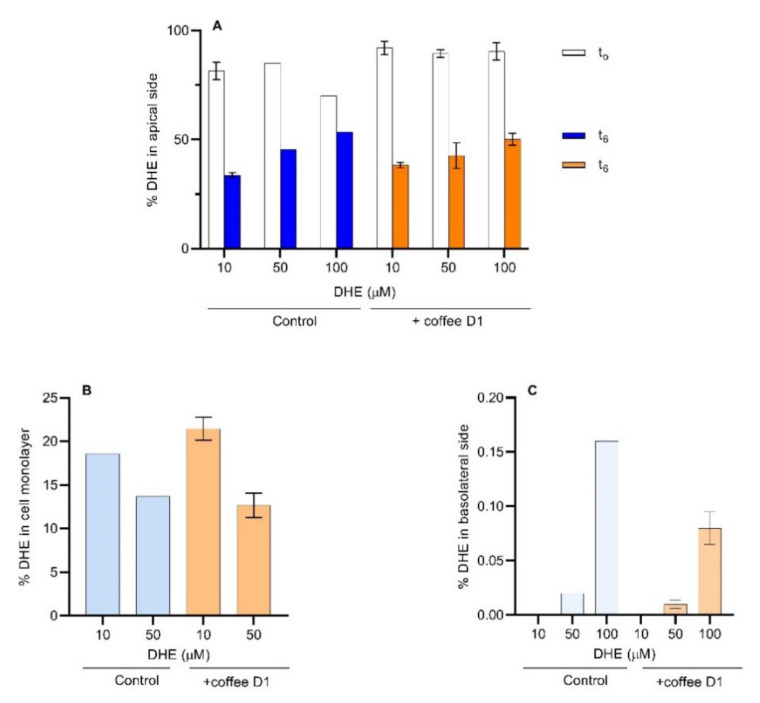
Assessment of the amount of DHE in the Caco-2 assay compartments. (**A**) The apical medium was analyzed at t_0_ (initial) and t_6_ (final). (**B**) The appearance of DHE in the basolateral medium was measured at t_6_. (**C**) The cellular sequestration of DHE was determined at t_6_. When the error bars are shown, n ≥ 2.

**Figure 7 foods-11-02935-f007:**
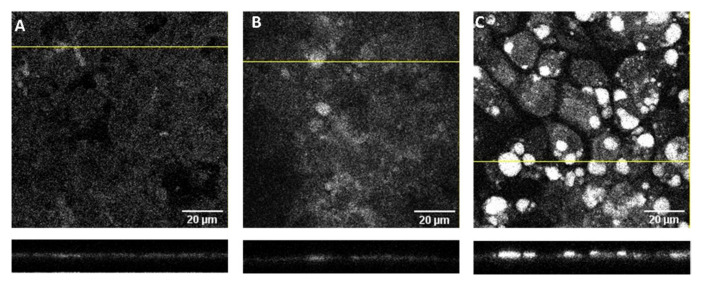
Confocal microscopy images of Caco-2 monolayers obtained with excitation light at 405 nm and emission detection from 408–490 nm. Representative optical section images along the *z*-axis (1 µm) are shown for (**A**) cell monolayers in the insert not used for any permeability assay and for cell monolayers exposed to DHE 100 μM (**B**) and both DHE 100 μM and coffee D1 (**C**) during a permeability assay of 6h. The lower insets show the localization of the fluorescence in the cross section of a z-x plane corresponding to the position of the yellow line.

**Table 1 foods-11-02935-t001:** Amount of LY transported and corresponding permeability coefficient in permeability assays through Caco-2 monolayers with a multi-time point assay with sampling at 2, 4 and 6 h. Effect of DHE (10 µM) emulsified by GCA (10 mM), in the absence and presence of coffee extracts, n = 2.

	% Transport of LY 0–2 h	% Transport of LY 2–4 h	% Transport of LY 4–6 h	*P*_app_ of LY (×10^−6^) ^a^
LY	1.6	0.9	1.3	0.64
DHE/GCA	0.84 ± 0.45	1.4 ± 0.20	0.81 ± 0.21	0.40 ± 0.09
Coffee extracts				
D1	0.41 ± 0.005	0.54 ± 0.26	0.60 ± 0.38	0.30 ± 0.19
D3	0.45 ± 0.12	0.46 ± 0.04	0.60 ± 0.37	0.30 ± 0.18
L1	0.51 ± 0.34	0.64 ± 0.14	0.85 ± 0.42	0.42 ± 0.21
L3	0.46 ± 0.07	0.19 ± 0.12	0.38 ± 0.02	0.19 ± 0.01

^a^ *P*_app_ value indicated is for the last time interval of 2 h. No statistically significant differences are observed between the samples.

**Table 2 foods-11-02935-t002:** Amount of LY transported and corresponding permeability coefficient in permeability assays through Caco-2 monolayers with a single time point assay of 6 h. Effect of increasing DHE concentrations (10, 50 and 100 µM) emulsified by GCA (10 mM), in the absence and presence of coffee extract D1, n = 2.

		% Transport of LY 6 h	*P*_app_ of LY (×10^−6^)
LY		1.6	0.27
	[µM]		
DHE/GCA	10	0.7	0.11
	50	1.1	0.18
	100	0.6	0.11
Coffee extract			
D1	10	0.66 ± 0.13	0.11 ± 0.02
	50	0.64 ± 0.05	0.11 ± 0.01
	100	0.40 ± 0.12	0.07 ± 0.02

No statistically significant differences are observed between the samples.

## Data Availability

Data will be sent to interested researchers by request to the corresponding authors.

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
