# Peer review of "Effect of Coffee on the Bioavailability of Sterols"

_foods, 2022, doi:10.3390/foods11192935_

Round 1

Reviewer 1 Report

The topic of the manuscript entitled "Effect of coffee on the bioavailability of sterols" is interesting, and the manuscript is well designed and presented. 

1. Please add some data details in the abstract;

2. The references format need to be revised

3. Data in the table or figures should be presented  as M+-SD or other else, and in the statistic analysis should give the details. 

Author Response

Reviewer 1

The topic of the manuscript entitled "Effect of coffee on the bioavailability of sterols" is interesting, and the manuscript is well designed and presented. 

  1. Please add some data details in the abstract;

The abstract was rephrased considering the reviewer comments, namely including the range of permeability coefficients obtained.

  1. The references format need to be revised

The format of all references was addressed and revised when needed.

  1. Data in the table or figures should be presented  as M+-SD or other else, and in the statistic analysis should give the details. 

Data is presented as average ± standard deviation in all conditions where replicates were performed. The number of replicates is now indicated in the figures caption. The statistical analysis performed is explained in section 2.10.

Reviewer 2 Report

Comments and Suggestions for Authors:

The authors explored the beneficial effect of coffee extracts on the improvement for the bioavailability of sterols. The topic fits the journal, and the story looks interesting. However, I have some questions about the design of this work.

1. Line 120: The coffee samples were prepared using expresso machine. Is the protocol (parameters, conditions, etc.) the same as or similar to daily process method, or with specific reasons? Since the authors have explained why use DHE/why use Caco-2/other materials, I suggest they can explain this point more clearly.

2. Using such exaction protocol, what kinds of constituents (the chemical basis) were the actual effective components. I suggest the authors to either add the related data (e.g., LC/MS-based profiling) or discuss this point as a limitation.

3. Line 132: The whole experiment was based on cell model. The definition of “bioavailability” is based on systemic circulation, which means in vivo investigation. Please the authors discuss this point objectively in an additional paragraph “limitation”.

4. Line 253: Is it a quantitative method? If so, please show methodology validation; otherwise, it should be a semi-quantitative method (even using HPLC). Besides, if it is a validated method, please show references.

5. Since the authors used four kinds of coffee extracts for this experiment, their comparison together with the possible explanation should be provided, which may be, again, related to the chemical characteristics. Therefore, an addition of comparative profiling and appropriate statistical analyses like correlation analysis are recommended.

6. There are not a little format errors and grammar mistakes throughout the whole text. Please revise them.

Author Response

Reviewer 2

The authors explored the beneficial effect of coffee extracts on the improvement for the bioavailability of sterols. The topic fits the journal, and the story looks interesting. However, I have some questions about the design of this work.

  1. Line 120: The coffee samples were prepared using expresso machine. Is the protocol (parameters, conditions, etc.) the same as or similar to daily process method, or with specific reasons? Since the authors have explained why use DHE/why use Caco-2/other materials, I suggest they can explain this point more clearly.

Considering the suggestion of the reviewer, the following statement was rephrased in section 2.2 “Arabica Brazil green grains were subjected to a commercial roasting procedure in the company FEB, originating commercial light (L) and dark (D) roasted coffee beans. Using a coffee grinder, coffee beans were milled with two different grinding levels to obtain fine (1) and coarse (3) particle sizes. The final coffee samples, denoted as L1 (light roast and fine grinding), D1, L3, D3 were extracted in an expresso machine (Flama 10) operating at 15 bar in the proportion of 6.0 g per 40 mL of water. In order to have uniform extractions, a tamper of 51 mm of diameter was used to pressure the coffee powder in the coffee portafilter basket previous to the extraction.”

  1. Using such exaction protocol, what kinds of constituents (the chemical basis) were the actual effective components. I suggest the authors to either add the related data (e.g., LC/MS-based profiling) or discuss this point as a limitation.

In the introduction section, discussion regarding the effect of roasting and gridding on chemical profile of coffee brews was included.

  1. Line 132: The whole experiment was based on cell model. The definition of “bioavailability” is based on systemic circulation, which means in vivo investigation. Please the authors discuss this point objectively in an additional paragraph “limitation”.

The reviewer is right concerning the definition of bioavailability. However, Caco-2 permeation studies are considered good in vitro models to infer in vivo bioavailability of poorly metabolized and slowly excreted compounds as shown in several examples in literature. One of the examples are the works of R. Glahn and other authors. To make clear the concept used in this work, changes in the abstract and introduction were done, considering the permeation of sterol through Caco-2 cell monolayers as a potential measure of sterol bioavailability. Additional references were added to the manuscript.

Glahn, R. The use of Caco-2 cells in defining nutrient bioavailability: Application to iron bioavailability of foods. In Designing Functional Foods: Measuring and Controlling Food Structure Breakdown and Nutrient Absorption; 2009; pp. 340–361 ISBN 9781845694326.

Tran, V. N.; Viktorova, J.; Ruml, T., Mycotoxins: Biotransformation and Bioavailability Assessment Using Caco-2 Cell Monolayer. Toxins 2020, 12.

Punt, A.; Peijnenburg, A.; Hoogenboom, R.; Bouwmeester, H., Non-Animal Approaches for Toxicokinetics in Risk Evaluations of Food Chemicals. ALTEX-Altern. Anim. Exp. 2017, 34, 501-514.

Yu, H. L.; Huang, Q. R., Improving the Oral Bioavailability of Curcumin Using Novel Organogel-Based Nanoemulsions. Journal of Agricultural and Food Chemistry 2012, 60, 5373-5379.

  1. Line 253: Is it a quantitative method? If so, please show methodology validation; otherwise, it should be a semi-quantitative method (even using HPLC). Besides, if it is a validated method, please show references.

The method was validated as indicated in the manuscript Supplementary Materials, sections S1 to S3. Three independent calibration curves were performed, covering the whole range of concentrations encountered in the samples analyzed. Reproducibility in the chromatograms for LY and DHE quantification (retention time and area), as well as recovery efficiency of LY and DHE extraction in the presence of coffee samples (apical compartment) and BSA (basolateral compartment) was also evaluated.

  1. Since the authors used four kinds of coffee extracts for this experiment, their comparison together with the possible explanation should be provided, which may be, again, related to the chemical characteristics. Therefore, an addition of comparative profiling and appropriate statistical analyses like correlation analysis are recommended.

The results were discussed taking in account the different coffees used. The following statement was included in the discussion section, complementing the information provided in the introduction:

“The roasting and grinding of coffee promote the extraction of polysaccharides [15,20]. Considering that these high molecular weight molecules may bind bile salts [9], this could be a reason by which sterol bioaccessibility is being affected differently by these coffee samples, although other compounds may also be responsible for this effect. “

  1. There are not a little format errors and grammar mistakes throughout the whole text. Please revise them.

The manuscript was thoroughly revised by the authors.

Reviewer 3 Report

This manuscript investigates the effect of coffee on the bioavailability of sterols, givin an interesting owerview.

However, some changes are necessary to increase the scientific soundness. 

Line 28/29-"I" means what?

Line 122 - what means FEB

L285 - what is meant by LY

Eq1. please explain all variables used in it, as well clarify waht are deltaQB and VA, mentioned in lines 291 & 293 but are not used in Eq1

Figures: use of "Ctrl" which is an key on the keybord is not appropriate 

In tables should be added signs or letters indicating statistical differences (see table 1 Papp for L1 and L3?!)

Also, use the same order in graphs and tables (in tables, samples L are displayed first, then D, while in graphs it's the other way around?!)

Supplementarx material:

Rewrite it - in Fig S1 can be added infos from Table S1, etc (Fig3 relate with data from Table S2)

Table S3-last column - it needs to be indicated why the SD is missing for the LY

Figure S5- why is not added the Coffee L3?

Sincerely,

Author Response

Reviewer 3

This manuscript investigates the effect of coffee on the bioavailability of sterols, givin an interesting owerview.

However, some changes are necessary to increase the scientific soundness. 

Line 28/29-"I" means what?

It was a typo, now corrected in the revised manuscript.

Line 122 - what means FEB

FEB is the name of the company that provided the coffees. This is now clearly indicated in the revised manuscript.

L285 - what is meant by LY

LY stands for Lucifer Yellow, it is now defined in the materials section.

Eq1. please explain all variables used in it, as well clarify waht are deltaQB and VA, mentioned in lines 291 & 293 but are not used in Eq1

The identification of the variables in the text was corrected to match the notation used in the equation.

Figures: use of "Ctrl" which is an key on the keybord is not appropriate 

“Ctrl” is the abbreviation of the control sample. This was defined in the text and figure 4. The identification of those samples was changed to “control” to improve clarity.

In tables should be added signs or letters indicating statistical differences (see table 1 Papp for L1 and L3?!)

When not indicated by *, no statistically differences were observed between the samples. This is now indicated in table 1 and 2.

Also, use the same order in graphs and tables (in tables, samples L are displayed first, then D, while in graphs it's the other way around?!)

Table 1 was changed in accordance.

Supplementarx material:

Rewrite it - in Fig S1 can be added infos from Table S1, etc (Fig3 relate with data from Table S2)

The suggestion was followed in the revised manuscript.

Table S3-last column - it needs to be indicated why the SD is missing for the LY

Permeability of LY was extensively characterized by us in a previous publication (Pires et.al. Pharmaceutics 2021, 13, 1563) where it was found that for monolayers at day 22 post-seeding the average amount of LY that permeated during the first sampling time point (up to 1 h) was 0.3 % with CI95% [0.13, 0.71], and it was 0.23 % with CI95% [0.11, 0.50] on day 25 post-seeding, with a large variability observed for independent cell monolayers. Only one experiment with LY alone was performed for each independent cell monolayer preparation used in the present work. The values obtained were within the expected range. 

Figure S5- why is not added the Coffee L3?

This data was not added to Figure S5 because it is provided in the main manuscript. It was added to the revised SI to facilitate the comparison between the different coffee samples.

Round 2

Reviewer 1 Report

I have no questions.

Author Response

The authors acknowledge the reviewer for the revison of the manuscript.

Reviewer 2 Report

The authors responded appropriately and provided the required information. Now, the basis of the experiment, the design of the in vitro model, and the interpretation of the results, are obviously improved. No more comments from me.

Author Response

(The authors gave the same response as above.)
